# Hollow Spherical Pd/CdS/NiS with Carrier Spatial Separation for Photocatalytic Hydrogen Generation

**DOI:** 10.3390/nano13081326

**Published:** 2023-04-10

**Authors:** Xiao Wang, Fei Zhao, Nan Zhang, Wenli Wu, Yuhua Wang

**Affiliations:** 1School of Materials and Energy, Lanzhou University, Lanzhou 730000, China; 2National and Local Joint Engineering Laboratory for Optical Conversion Materials and Technology, Lanzhou University, Lanzhou 730000, China

**Keywords:** hollow spherical structure, spatial separation, co-catalyst, photocatalytic hydrogen evolution

## Abstract

Inspired by the unique properties of the three-dimensional hollow nanostructures in the field of photocatalysis, as well as the combination of co-catalyst, porous hollow spherical Pd/CdS/NiS photocatalysts are prepared by stepwise synthesis. The results show that the Schottky junction between Pd and CdS accelerates the transport of photogenerated electrons, while a p-n junction between NiS and CdS traps the photogenerated holes. As co-catalysts, the Pd nanoparticles and the NiS are loaded inside and outside the hollow CdS shell layer, respectively, which combines with the particular characteristic of the hollow structure, resulting in a spatial carrier separation effect. Under the synergy of the dual co-catalyst loading and hollow structure, the Pd/CdS/NiS has favorable stability. Its H_2_ production under visible light is significantly increased to 3804.6 μmol/g/h, representing 33.4 times more than that of pure CdS. The apparent quantum efficiency is 0.24% at 420 nm. A feasible bridge for the development of efficient photocatalysts is offered by this work.

## 1. Introduction

Ascribed to its high calorific potential with a pollution-free combustion process, hydrogen has been paid more and more attention around the world. It is considered one of the most suitable energy sources to substitute non-renewable energy [1,2,3]. Therefore, as a kind of green chemical technology, photocatalytic hydrogen evolution technology has broad application prospects and unique advantages in alleviating environmental pollution and energy crisis, receiving wide attention from researchers [4,5,6]. However, the existing photocatalyst is limited by the rapid recombination of photogenerated electrons and holes and the slow surface reaction process, which leads to the unsatisfactory performance of photocatalytic hydrogen production [7,8].

A complete photocatalytic H_2_ evolution process is dependent on many factors in the process, such as photo-absorption capacity, charge separation and transport efficiency, and photoelectron utilization of H_2_ generation [9,10,11]. In order to design an efficient photocatalyst, these factors need to be considered and optimized comprehensively. For this reason, researchers have proposed various strategies to improve the performance of semiconductor photocatalysts, including band structure engineering [12,13], nanostructure engineering [14,15], co-catalyst loading [16,17], surface and interface engineering [18,19], doping [20,21,22], Vacancy engineering [23,24], etc. Among them, the effectiveness and feasibility of nanostructural engineering have proved to be one of the most promising strategies [25,26]. The three-dimensional hollow structure has several peculiar properties in photocatalytic [27]. The hollow structure not only offers a significant specific surface area for surface reactions, hence boosting their rate, but it also promotes the scattering of light within the hollow structure, thereby enhancing photo-absorption capabilities [28,29]. Furthermore, a thinner shell can effectively decrease the charge transfer distance to the semiconductor surface, resulting in an accelerated charge transfer rate [30]. Zhang et al. [31] reported a medium empty tubular CoO_x_/TiO_2_/Pt photocatalyst, proving that the hollow structure provides a large specific surface area, which is favorable for the photocatalyst hydrogen generation reaction. Qiu et al. [32] compared the hydrogen production performance of solid and hollow nanostructure photocatalyst Co_9_S_8_/CdS and found that the hollow nanostructure had a better photocatalytic hydrogen evolution performance.

At present, the CdS in metal sulfide is a strong competitor in hydrogen evolution thanks to the suitable band gap (2.4 eV) and the easily controlled morphology [33,34,35,36]. However, a single CdS semiconductor still does not have a high photocatalytic hydrogen production performance, which is explained by the fast reorganization of electron-hole pairs as well as surface reactions that are too slow to consume charge efficiently [37,38,39]. The co-catalysts can effectively promote the transfer of photogenerated carriers to the surface active sites and increase the reaction rate while restraining the recombination [40,41]. Nevertheless, the loading of a single co-catalyst can only improve an electron or the hole migration efficiency, which has a very limited effect on promoting charge separation [42,43]. An approach to overcome this limitation is by dual co-catalyst loading to achieve simultaneous electron and ensure that the hole transfer has emerged. Zhang et al. [44] reported a novel MnOx/CdS/CoP photocatalyst in which the MnOx and CoP nanoparticles were used as reduction and oxidation co-catalysts, respectively. The loading of the dual co-catalysts not only provided abundant active sites but also accelerated the separation of photo-generated carriers, which enabled the photocatalyst to exhibit excellent hydrogen production performance. Moreover, combined with the construction of hollow morphology, it can effectively control the different reactions inside and outside the shell layer that achieves spatial carrier separation [45,46,47]. Loading the co-catalysts inside the shell layer is a dilemma due to the difficulty in controlling the loading process, while loading the co-catalysts outside the shell layer often requires complex reactions, which can also have an impact on the photocatalyst. Therefore, it is challenging to achieve precise, simple, and effective control of co-catalyst loading.

As a reduction co-catalyst, the density of the state of Pd near the Fermi level is higher than that of Pt, Au, and Ag, which is about 0.2 eV higher than that of Pt. As a result, Pd has a lower electron affinity compared to other precious metals and, thus, has a stronger electron capture capability, which may make it easier for electrons to transfer from Pd to the semiconductor surface [48,49]. The NiS of p-type semiconductors has also been extensively studied as an excellent oxidation co-catalyst [50].

Inspired by previous work, we designed a hollow spherical 0.7%-Pd/CdS/NiS-3% (P/CS/NS) photocatalyst to be used for high-efficiency photocatalytic hydrogen generation with visible-light irradiation. Through the process of amination, the Pd nanoparticles are accurately and firmly embedded within the CdS shell layer. On the other hand, the growth of the NiS outside the CdS shell layer is facilitated by co-precipitation in a straightforward and effective manner. In this study, we investigated the mechanism of loading duplex catalysts on both sides of the CdS shell layer to construct Schottky junctions and p-n junctions, respectively, to enhance carrier separation. This work broadens the way for designing other efficient photocatalysts with spatial separation effects.

## 2. Experimental Details

### 2.1. Materials

Ammonia (NH_3_·H_2_O, 25–28%), tetraethoxysilane (TEOS), sodium sulfide nonahydrate (Na_2_S), sodium sulfite anhydrous (Na_2_SO_3_), ethanol, isopropyl alcohol, Palladium(II) chloride (PdCl_2_), (3-aminopropyl) triethoxysilane (APTES), cadmium acetate (Cd(Ac)_2_), nickel nitrate(Ni(NO_3_)_2_), sodium citrate, thiourea (CH_4_N_2_S), and sodium hydroxide (NaOH) were purchased from Shanghai Macklin Biochemical Co., Ltd. (Shanghai, China), and deionized water (DI) was produced from Millipore (Milli-Q, 18.2 MΩ/cm; Millipore, Bedford, MA, USA). All the chemical reagents were analytical grade and used as received without any further purification.

### 2.2. Synthesis

*Synthesis of SiO_2_.* We synthesized silica nanospheres using the Stóber method (Figure 1). A total of 20 mL of anhydrous ethanol and 10 mL of ethyl orthosilicate (TEOS) were prepared as liquid A. In total, 40 mL of anhydrous ethanol and 120 mL of ammonia were used as liquid B. Mixing liquid A with liquid B. The SiO_2_ nanospheres in the suspension were separated by centrifugation, rinsed repeatedly, and dried in air at 60 °C.

*Synthesis of SiO_2_-NH_2_.* First, 0.6 g of the prepared SiO_2_ spheres were dispersed in 150 mL of isopropanol and sonicated for 30 min to make them uniformly dispersed, and then 3 mL of (3-aminopropyl) triethoxysilane (APTES) was added and reacted at 78 °C for 3 h. The obtained SiO_2_-NH_2_ samples were centrifuged with ethanol, washed repeatedly, and finally dried at 60 °C in air.

*Synthesis of SiO_2_@Pd.* The SiO_2_-NH_2_@Pd was prepared by the adsorption of heavy metals by amination. A total of 0.4 g prepared SiO_2_-NH_2_ spheres were dispersed into 60 mL of anhydrous ethanol and ultrasonicated for 30 min. The calculated mass of chloropalladium acid solution (10.05 mmol/L) was added, and then the diluted hydrazine hydrate solution was added drop by drop until it became discolored, so that Pd nanoparticles were uniformly deposited on SiO_2_-NH_2_. The obtained product was centrifuged, washed, and dried overnight. By changing the amount of chloropalladium acid solution added, the samples loaded with a different concentration gradient of Pd were prepared.

*Synthesis of SiO_2_@Pd/CdS.* We took 0.3 g of already prepared SiO_2_@Pd sample, disperse it in 150 mL of deionized water, then add 0.4 g of sodium citrate and ultrasonically disperse it for 30 min. Then 0.6 g cadmium acetate, 10 mL ammonia, and 0.33 g thiourea were added in turn, and the reaction was stirred at 60 °C with reflux for 3 h. The product was centrifuged, washed, and dried in air at 60 °C.

*Synthesis of SiO_2_@Pd/CdS/NiS.* In total, a 0.2 g sample of Pd/CdS was dispersed into 100 mL of deionized water, 0.026 mmol of Na_2_S was added, stirred until it was uniformly dissolved, and then the equal molar mass of nickel nitrate hexahydrate was added. Stir for 1 h. The product was washed and dried a under vacuum. Different loading amounts of SiO_2_@Pd/CdS/NiS were prepared by varying the amounts of added nickel and sulfur sources. Pure NiS is obtained by not adding SiO_2_@Pd/CdS.

*Synthesis of hollow Pd/CdS/NiS.* The sum of 0.2 g of SiO_2_@Pd/CdS/NiS was etched in a 5M NaOH solution at 80 °C for 3 h to remove the SiO_2_ template and obtain hollow spherical structures. The final product was collected by centrifugation, washed with water and anhydrous ethanol, and dried at 60 °C under air. The CdS/NiS samples were prepared by not adding Pd. The optimal samples used for characterization and testing in the paper were 0.7%-Pd/CdS/NiS-3% (P/CS/NS).

### 2.3. Characterization

The powder crystal structure of the samples was characterized by a Bruker D2 PHASER X-ray diffractometer (XRD, Bruker Inc., Billerica, MA, USA) using Cu Kα radiation (λ = 1.54184 Å). The chemical composition was detected by X-ray photoelectron spectroscopy (XPS, PHI-5702, Physical Electronics Inc., Chanhassen, MN, USA) under Al Ka irradiation. The morphologies of the prepared samples were detected by field emission scanning electron microscopy (FESEM, Thermo Fisher Scientific Inc., Apreo S, Bedford, MA, USA). Moreover, the transmission electron microscopy (TEM), the high-resolution transmission electron microscopy (HRTEM), the scanning transmission electron microscopy (STEM), and the EDX elemental scanning are tested via the F30 S-TWIN electron microscope (Tecnai G2, FEI Company, Hillsboro, OR, USA). The absorption spectra and transmittance spectra are obtained by a UV–Vis-NIR spectrophotometer (PE Lambda 950, Norwalk Inc., Norwalk, CT, USA) with BaSO_4_ as a reference with wavelengths in the 250–1300 nm range. The PL emission spectra were obtained by an FLS-920T fluorescence spectrophotometer (HORIBA Jobin Yvon Inc., Paris, France) equipped with a 450 W Xe light source. The electrochemical and photoelectrochemical properties of the samples were tested using the CS310 electrochemical workstation of Crestor Instruments Co. (Wuhan, China).

### 2.4. Photocatalytic Activity

The photocatalytic H_2_ evolution activity measures of the samples were carried out using an automatic gas circulation system (GC-2014C, PerfectLight Labsolar 6A, PerfectLight Co., Beijing, China). In a typical photocatalytic hydrogen evolution test, 25 mg of photocatalyst powder was dispersed in 0.1 M Na_2_S and Na_2_SO_3_ solutions, and the entire gas circulation system, including the reaction unit, was evacuated for 30 min to remove air completely, and then the reaction unit was kept at 6 °C using a cold trap. Then the reaction unit was irradiated under a 300 W Xe lamp equipped with a UV-IR-CUT filter to obtain visible light (420–780 nm). The intensity of the light source was 141 mW/cm^2^ and the distance between the light source and the reactor was 17 cm. Ultimately, the H_2_ evolution was investigated through an online gas chromatograph (GC-7920, PerfectLight Co., Beijing, China) equipped with a thermal conductivity detector (TCD), using N_2_ as carrier gas and a 5 Å molecular sieve column.

The photocatalyst stability test method is the same as the hydrogen production activity test conditions, with the difference that one sample is tested for hydrogen production five times consecutively without the addition of sacrificial agents, and the stability of the photocatalyst is measured by the change in the amount of hydrogen produced five times.

The apparent quantum yields (AQYs) for H_2_ production were determined by a single-wavelength, LED light source, photochemical reactor (CEL-PCRD300-12) with gas chromatography (GC-7920). The test method was the same as the catalytic hydrogen production measurement. The optical power densities of the incident light are measured with an irradiatometer (CEL-NP2000-2). The AQYs are calculated according to Equation (1).
(1)$$AQY\left(\%\right)=\frac{number\;of\;evolved\;H_{2}\;molecules\;\times \;2}{number\;of\;incident\;photons}\times 100\%$$

## 3. Results and Discussion

### 3.1. Characterizations of As-Prepared Photocatalysts

X-ray diffraction (XRD) is utilized to investigate the physical phase structure of the samples that have been prepared. Specifically, the prepared sample is greenockite CdS (JCPDS NO. 41-1049) and presents evident diffraction peaks that correspond to the crystal planes (100), (002), (101), (110), (103), and (201), respectively. These diffraction peaks align notably well with the standard card of CdS (Figure 2) [15]. It is noteworthy that the diffraction peaks of CdS display a significant broadening, which can be attributed to the small grain size of the prepared CdS samples. The absence of other peaks indicates that no impurities are generated. Comparing the XRD patterns after loading different co-catalysts, it can be found that the loading of co-catalysts does not influence the crystal structure of CdS. The composite sample does not display distinct characteristic peaks for Pd and NiS. This observation can be attributed to the inadequate loading of Pd and NiS on the shell layer of the sample, resulting in the inability to visualize their corresponding diffraction peaks (the contents of Pd and NiS are confirmed by ICP-OES, Appendix A). The characteristic diffraction peaks of pure NiS indicate that it is prepared as a hexagonal NiS (JCPDS NO. 77-1624). Therefore, a further demonstration of the presence of Pd and NiS is required subsequently.

The SEM is utilized to document the morphological and structural modifications of the samples throughout the procedure. As shown in Appendix A, the prepared SiO_2_ templates exhibit a perfect spherical morphology, with a smooth surface, and no impurities present, and the spheres are equal in size with a diameter of around 500 nm. When after amination, the Pd is successfully captured at surfaces of SiO_2_ nanospheres, the bright spots in the figure can be Pd nanoparticles (Appendix A). The CdS produced through the solvent heat method is evenly coated onto the smooth Pd@SiO_2_. Subsequently, the NiS co-catalyst is loaded via the co-precipitation method, resulting in the morphological structure in Appendix A. Upon removal of the SiO_2_ nanospheres by alkaline etching in the NaOH solution, the Pd/CdS/NiS photocatalyst with a hollow structure is formed (Appendix A).

The morphological and structural information of samples from the TEM is further characterized. As shown in Figure 3a–d, it corresponds to the SEM figures one by one. It can be clearly observed that the CdS forms a shell structure with the internal SiO_2_ template spheres (Figure 3c). When the hollow spherical Pd/CdS/NiS sample is obtained after removing the SiO_2_ template, its morphology is in Figure 3d, and it is noticed that the shell layer of the Pd/CdS/NiS sample remains stable after the process of alkaline etching. This indicates that the alkaline etching method is feasible and does not lead to any deformation or breakage of the shell layer structure.

To explore the more detailed morphology at the shell layer, the morphology at the shell layer is carried with a stepwise enlargement, as displayed in Figure 3e,f. It can be identified that the hollow spherical Pd/CdS/NiS has a thin shell layer with a thickness of about 25 nm, which is favorable for charge transport. Then, a high-resolution TEM analysis of the shell layer part can observe the presence of lattice stripes with a spacing of 0.298 nm, corresponding to the (100) grain surface of NiS [51], indicating that the NiS is effectively trapped in the outer shell layer of the hollow spherical CdS (Figure 3g). We select a partially broken sample to investigate the interior of the hollow structure and the loading of the Pd nanoparticles (Figure 3h). Numerous mesopores with diameters between 3 nm and 5 nm can be observed at the inner shell layer of the CdS, suggesting that hollow Pd/CdS/NiS samples have a mesoporous structure (Figure 3i). HRTEM characterization of the inner shell layer is performed, and the lattice stripes at a spacing of 0.22 nm correspond to the (111) crystal plane of the Pd nanoparticles, while clear lattice stripes of CdS can be observed with a spacing of 0.33 nm and 0.23 nm corresponding to the (002) and (101) crystal planes of CdS (Figure 3j), respectively. At the same time, combined with the experimental process of this work, it is demonstrated that the Pd particles are effectively captured in the inner shell layer of the hollow CdS spheres, as well as the dispersion is relatively dispersed and firmly bound with the CdS shell. This tight binding facilitates the transport of photogenerated carriers, reducing the energy loss in the photocatalytic process. The elemental distribution of the Pd/CdS/NiS is determined by the element mapping method (Figure 3k). The mapping spectra clearly show the existence of Cd, S, Ni, and Pd in the Pd/CdS/NiS, while the elemental distribution corresponds to the sample morphology, affirming the successful synthesis of the Pd/CdS/NiS samples. Further evidence of this is provided by the EDS line scan. It can be noticed that at the location of the Pd/CdS/NiS shell layer, the EDS intensity of the Pd starts to decrease, first after enhancing near the inner side of the shell layer, and the EDS intensity of Ni starts to fall only after enhancing near the outer side of the shell layer. This indicates that Pd is more located on the inner side of the Pd/CdS/NiS shell layer, while NiS is more located on the outer side. (Appendix A).

The surface elemental composition and chemical state of the samples are analyzed by XPS. It can be concluded that each sample contains its corresponding element from the full spectra scans of the XPS of the CdS and Pd/CdS/NiS (Appendix A), agreeing with the EDX results. The high-resolution XPS spectrum of Cd 3d in Figure 4a shows that the electronic binding energies of Cd 3d_5/2_ and Cd 3d_3/2_ are 404.69 eV and 411.44 eV, respectively, which is consilient with the Cd-S bond energy of Cd^3+^. Two peaks were obtained for S 2p_3/2_ and S 2p_1/2_ at a binding energy of 161.35 eV and 162.50 eV, corresponding to S^2+^ (Figure 4b) [52]. Remarkably, comparing the Cd 3d of the original sample CdS with the high-resolution XPS, the Cd 3d in the Pd/CdS/NiS sample changes toward a lower binding energy, while the binding energy of S 2p shows a similar trend, which indicates that the binding of the dual co-catalyst to CdS can influence the electronic interaction and, thus, naturally establish an effective bridge for charge transfer [28]. It can be determined that the peaks appearing at 853.19 eV and 870.87 eV correspond to Ni 2p_3/2_ and Ni 2p_1/2_ (Figure 4c), respectively. This finding aligns with the previous literature that reported the same correlation for NiS [51], effectively demonstrating the successful precipitation of NiS in the external shell layer of the hollow CdS. Next, we explore the state of the Pd element, the peaks of Pd 3d_5/2_ and Pd 3d_3/2_ at 335.45 eV and 340.52 eV are identified in the Pd/CdS/NiS sample as shown in Figure 4d, signifying a valence state of 0 for the Pd element, proving that the loaded Pd element is a Pd nanoparticle [34]. While the characteristic peaks of Ni 2p and Pd 3d in the figure show weak peaks, which is due to the smaller amount of loading pairs.

### 3.2. Photocatalytic H_2_ Evolution Activity Measurements

The photocatalytic hydrogen evolution effect of the prepared catalysts is evaluated with Na_2_S and Na_2_SO_3_ solutions (hole scavenger) by visible light when irradiated. The corresponding test data are summarized in Figure 5. The H_2_ generation of the original hollow spherical CdS was still very unsatisfactory (113.8 μmol/g/h) after 4 h irradiation, which could be ascribed to the low photocatalytic hydrogen generation performance resulting from the faster restructuring of electrons with holes in the CdS semiconductor (Figure 5a). The H_2_ generation of the CdS/NiS photocatalyst started to increase with the addition of co-catalyst NiS, which meant that the loading of the NiS oxidation co-catalyst could effectively increase the photocatalytic hydrogen production activity of CdS. It was found that the optimal activity of the CdS/NiS samples was achieved when the NiS loading capacity was 3%. Next, on the basis of the optimal loading of NiS, we explored the optimal loading of Pd nanoparticles (Figure 5b). The hydrogen production of the Pd/CdS/NiS samples substantially increased with the level of Pd nanoparticle loading, indicating that Pd nanoparticles with NiS as the reduction and oxidation co-catalysts of the Pd/CdS/NiS samples, respectively, play very significant roles in the surface charges migration as well as electron-hole separation ability of Pd/CdS/NiS. We also could find that the hydrogen production of Pd/CdS/NiS decreased partially when the loading of the dual co-catalyst was more extensive, it was due to the increase of the loading density of the dual co-catalyst, which caused the coverage of the reaction sites of Pd/CdS/NiS, resulting in a decreased contact of reactants with the catalyst, thus leading to a decrease of the photocatalytic hydrogen precipitation performance. When the loading of the dual co-catalyst was lower, it could not generate enough effective active sites for photocatalytic hydrogen production. As a result, it could be inferred that the optimum combination of 0.7% Pd and 3% NiS was most suitable for boosting the H_2_ generation of the CdS. The hydrogen production rate of 0.7%-Pd/CdS/NiS-3% photocatalyst reached 3804.6 μmol/g/h (Appendix A), as compared to the pure CdS by 33.4 times, indicating excellent photocatalytic H_2_ generation activity (Appendix A). The hydrogen production efficiency of the Pd/CdS/NiS was analyzed in Figure 5c by measuring its response to varying wavelengths. A bandpass filter was installed to generate monochromatic incident light during the photocatalytic hydrogen production process. The photocatalytic AQY values of Pd/CdS/NiS at wavelengths 365, 400, 420, 450, 500, and 550 nm were 5.7‰, 2.28‰, 2.24‰, 2.86‰, 1.82‰, and 0.5‰, respectively. The Pd/CdS/NiS sample exhibits no photocatalytic activity for hydrogen evolution when the incident light wave is larger than 600 nm. A tendency that agreed well with the absorption spectrum of hollow CdS suggested that the production of photogenerated carriers in the Pd/CdS/NiS scheme depended on the CdS. Besides excellent photocatalytic hydrogen production efficiency, stabilization of recovery was also a key criterion for successful application in photocatalysts (Figure 5d). The H_2_ evolution activity of the original CdS photocatalyst decreased by 35% in the five-cycle tests, which was owed to the low hydrogen generation efficiency caused by the photo-corrosion of the CdS photocatalyst. The hydrogen production of the Pd/CdS/NiS sample began to decrease after the first cycle. After the second cycle, its hydrogen generation exhibited a slight upward trend, which could be due to the fact that a very small part of Pd atoms on the CdS shell reacted with the S element produced by the sacrifice agent in solvent to generate PdS during the 8 h test process, and PdS could further promote the photocatalytic H_2_ evolution efficiency for CdS [15,34]. As a result, there was a slight trend of improvement in subsequent cycle tests. After five cycles, the Pd/CdS/NiS still maintained the high efficiency of hydrogen production performance, which indicated that the hollow spherical Pd/CdS/NiS photocatalyst had high stability.

### 3.3. Photophysical and Electrochemical Properties

The UV-Vis DRS can reflect the capacity of a material to absorb light at the corresponding wavelength. As shown in Figure 6a, at approximately 542 nm, the pristine CdS exhibits a discernible absorption edge, which is dictated by the intrinsic band gap width of the material. With the incorporation of the NiS co-catalyst, the CdS/NiS composite displays a remarkable increase in absorption within the 500 to 800 nm range, suggesting that the presence of the NiS co-catalyst can effectively enhance the visible light absorption capacity of CdS. With the loading of the Pd nanoparticles, the absorption edge is unchanged, which reflects a weak effect for CdS/NiS lattice [50]. The phenomenon illustrates that the dual co-catalyst is effective in enhancing the photo-absorption of the hollow CdS. Carrier dynamics can be studied by using photoluminescence spectroscopy (PL), and the carrier separation between samples can be compared by comparing the PL strength of materials. With regards to the band gap emission, an emission band around 542 nm can be observed, which is believed to result from the recombination of electrons with holes in the CdS. For the CdS/NiS samples, their PL intensity decreases significantly, indicating that the inclusion of NiS can effectively mitigate this shortcoming of the CdS. Among all of the samples, the Pd/CdS/NiS sample exhibits the weakest PL intensity (Figure 6b), which reveals that the Pd and NiS co-catalyst can make the reorganization possibility of the photogenerated carriers much weaker, thus the photocatalytic H_2_ generation performance is greatly improved.

The transient photocurrent response is adopted to study the charge transport motion mechanics for different samples. When a light source is turned on, the photocurrent density of the photocatalyst increases dramatically and then remains stabilized, representing that the photocatalyst is highly responsive to visible light. Upon light turn-on, a spike emerges in the CdS, indicating the transient buildup of the charge brought about by light stimulation. This serves as evidence that the CdS generates a substantial quantity of efficient carriers, rather than recombination. With the loading of the NiS co-catalyst, the photocurrent of the CdS/NiS samples increases, signifying the generation of more available photogenerated carriers. Remarkably, the photocurrent density of the Pd/CdS/NiS sample is the highest. Compared with the original CdS, the higher photocurrent density of Pd/CdS/NiS means the best optical response and the easiest carrier separation capability (Figure 6c). Furthermore, various samples are tested via the EIS to assess the surface photoelectronic transfer ability of the photocatalyst (Figure 6d). In the Nyquist figure, the small diameter of the half-circle means that the resistance to charge transfer on the sample surface is weak, which certifies that photoelectrons can transfer quickly on the surface of the catalyst. The radius represented by the Pd/CdS/NiS sample is still the smallest among all the samples tested, which resembles this result for the transient photocurrent response. This characterization again confirms that loading dual co-catalyst can effectively improve the surface charge transport ability of the sample.

### 3.4. Photocatalytic Mechanism Analysis

In order to analyze the mechanism of the Pd/CdS/NiS samples in photocatalytic reactions, we test the Mott-Schottky (M-S) curves of pure CdS and NiS and relate their UV-Vis DRS to calculate the band gap, conduction band (CB), and valence band (VB) positions of the samples. As displayed in Figure 7a, CdS has a visible light response with a band gap of 2.34 eV through the DRS of CdS. The band gap of NiS is presented in Figure 7b, with a value of 1.32 eV, which indicates that the NiS has a strong light absorption ability. The Pd/CdS/NiS has a band gap of 2.21 eV (Appendix A). Moreover, as an auxiliary catalyst, the NiS can enhance the photo-absorption ability of the CdS, as demonstrated by the DRS results of the Pd/CdS/NiS. To obtain further band positions, the Mott-Schottky (M-S) curves of CdS and NiS are collected at AC frequencies of 0.5, 1.0, and 1.5 kHz. By extending the linear part of the M-S diagrams in Figure 7c,d, the flat band potentials (E_fb_) of CdS and NiS can be determined as −0.66 and 1.29 V (V vs. Ag/AgCl). Notably, the positive slope of the M-S curves of CdS means that CdS is the n-type semiconductor, while in NiS, the negative slope means that this is the p-type semiconductor. According to the literature, the conduction band potential (E_CB_) of the n-type semiconductors and the valence band potential (E_VB_) of the p-type semiconductors are consistent with their flat band potentials [10,53]. Therefore, the CB of CdS is evaluated as −0.66 eV, while the VB of NiS is 1.29 eV. Based on the formula of valence band potential: E_VB_ = E_CB_ + Eg, it can be estimated that the E_VB_ of CdS is 1.68 eV, while the E_CB_ of NiS is −0.03 eV.

In view of the previous analysis, a mechanism of photocatalytic hydrogen generation by Pd/CdS/NiS is proposed. As shown in Figure 8, under the continuous irradiation of visible light, electrons in the valence band of CdS transition to the conduction band, leaving holes in the valence band. When the electrons are transferred to the surface of CdS, the Pd nanoparticles act as a reduction co-catalyst to form a Schottky junction with CdS, allowing the electrons to migrate rapidly to the inner surface-loaded Pd nanoparticles for hydrogen reduction reactions, preventing the rapid compounding of photogenerated electron-hole pairs. Since NiS is the p-type semiconductor and CdS is the n-type semiconductor, when the NiS is loaded on the outer shell layer of the CdS, a p-n junction is formed thanks to their different Fermi energy levels [50]. As a result of the existence of a built-in electric field in this p-n junction, the photogenerated holes in the CdS can be transferred to the NiS and removed by the hole sacrificial agent. The LSPR effect between CdS and Pd accelerates the transport of photogenerated electrons. Meanwhile, a p-n junction between the CdS and NiS assists in the transfer of photogenerated holes from the CdS to NiS. Electrons and holes form an optimal spatial separation effect.

## 4. Conclusions

In summary, through a careful and rational consideration of morphological design and co-catalyst loading, a novel hollow spherical Pd/CdS/NiS photocatalyst is developed. For the photocatalytic reaction, optimizing the three key influencing factors is essential, and this design achieves just that. Pd nanoparticles act as carriers for electrons, while the NiS acts as traps for holes, accelerating both the transfer and separation of the photogenerated carriers, which combine with the advantage of the hollow structure to form a carrier spatial separation effect. Compared with pure CdS, the Pd/CdS/NiS photocatalyst showed a significant enhancement in the photocatalytic H_2_ production activity and displayed outstanding stability. This approach can provide a reasonable pathway for exploiting spatially separated and efficient photocatalysts.

## Figures and Tables

**Figure 1 nanomaterials-13-01326-f001:**
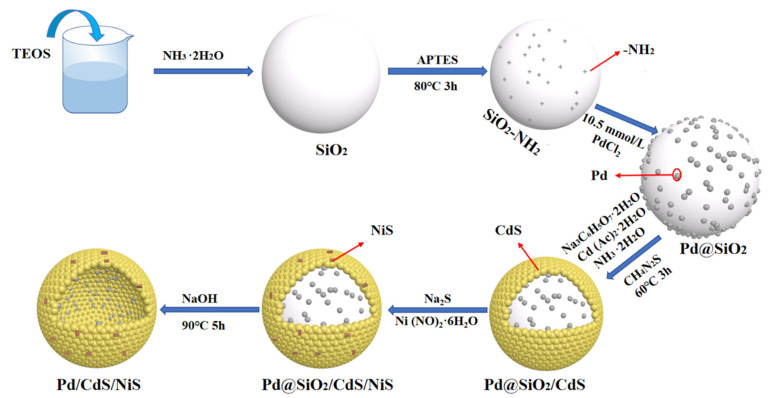
Schematic illustration of the synthesis process Pd/CdS/NiS photocatalyst.

**Figure 2 nanomaterials-13-01326-f002:**
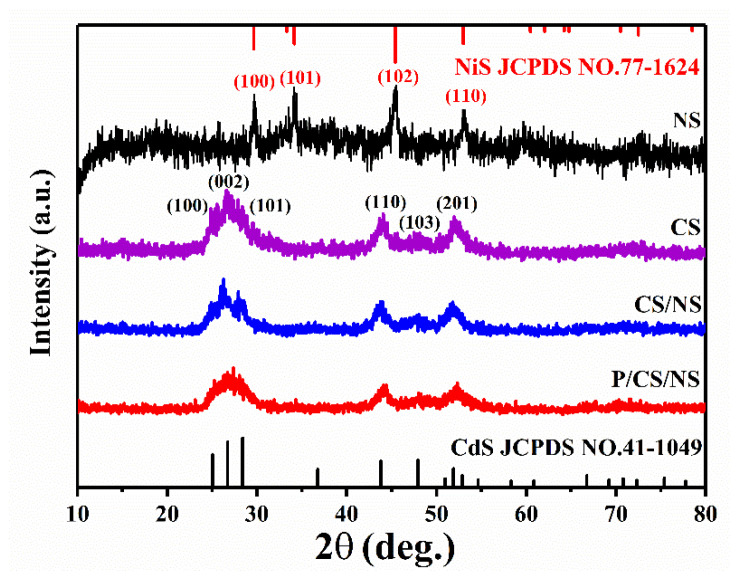
XRD patterns of NiS, CdS, CdS/NiS, and Pd/CdS/NiS.

**Figure 3 nanomaterials-13-01326-f003:**
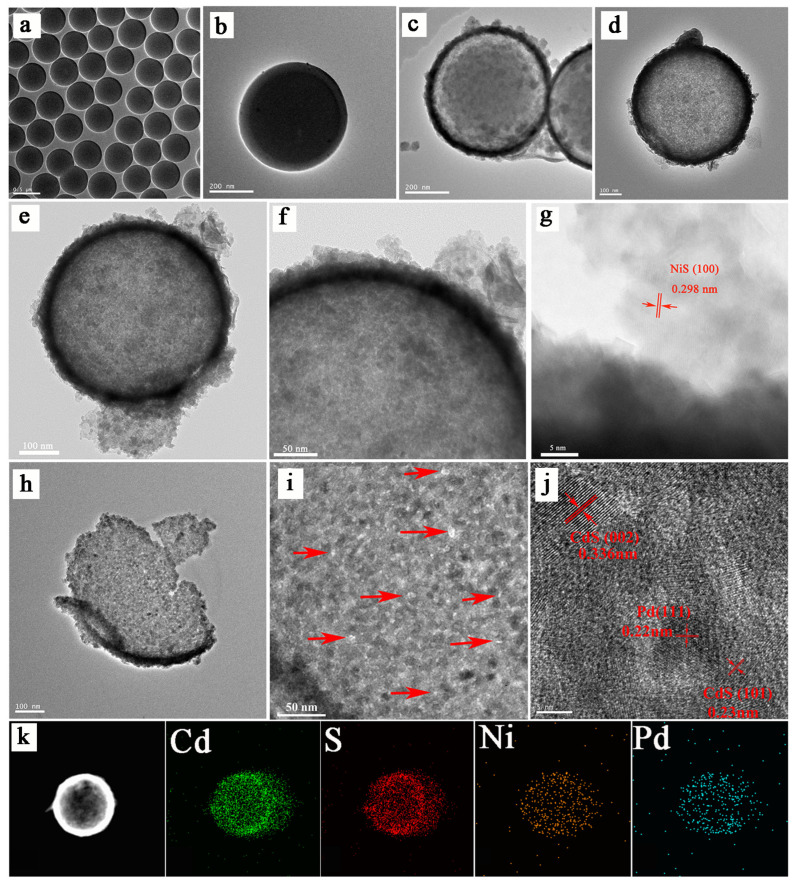
TEM figure of (**a**) SiO_2_; (**b**) SiO_2_@Pd; (**c**) SiO_2_@Pd/CdS/NiS; (**d**–**f**) hollow Pd/CdS/NiS; (**g**) High-resolution TEM image of NiS; (**h**) a broken Pd/CdS/NiS hollow sphere; (**i**) a partially magnified hollow sphere of broken Pd/CdS/NiS (Arrow marks the mesopore); (**j**) High-resolution TEM image of broken Pd/CdS/NiS hollow spheres; (**k**) EDX elemental mapping images of Cd, S, Ni, and Pd.

**Figure 4 nanomaterials-13-01326-f004:**
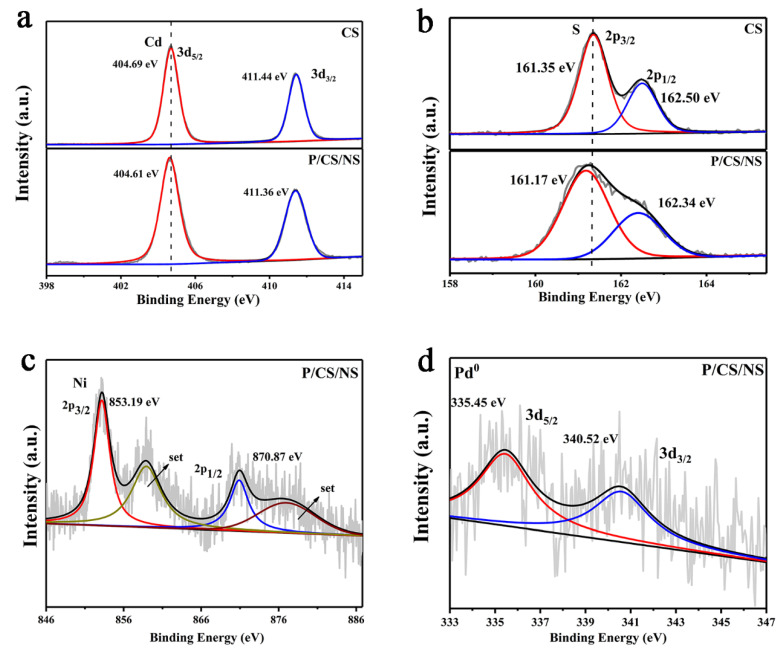
High-resolution XPS spectra of (**a**) Cd 3d; (**b**) S 2p in CdS and Pd/CdS/NiS, (**c**) Ni 2p; (**d**) Pd 3d in Pd/CdS/NiS.

**Figure 5 nanomaterials-13-01326-f005:**
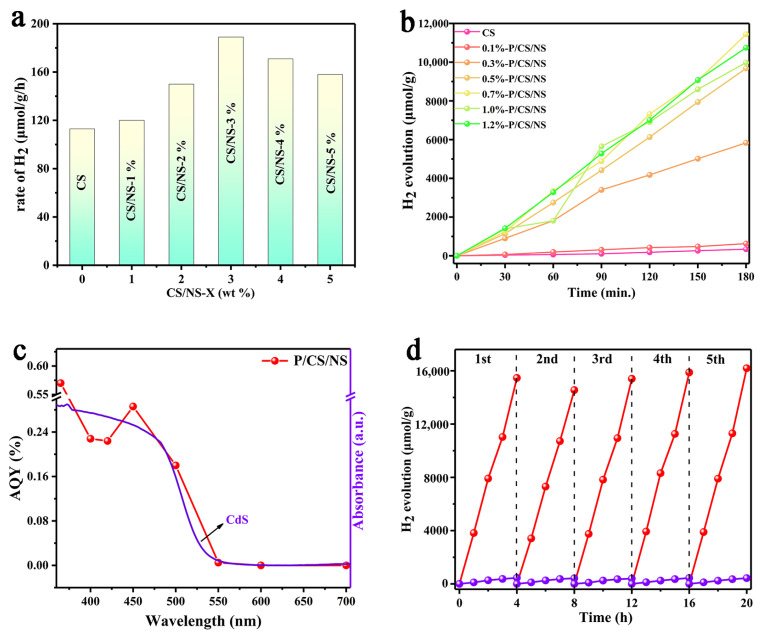
(**a**) H_2_ evolution of CdS/NiS loaded with different ratios of NiS after 4 h of light irradiation (λ ≥ 420 nm); (**b**) H_2_ production in Pd/CdS/NiS-3% loaded with different ratios of Pd (λ ≥ 420 nm); (**c**) The apparent quantum yield (AQY) of Pd/CdS/NiS; (**d**) Cycling stability test of Pd/CdS/NiS and CdS (λ ≥ 420 nm).

**Figure 6 nanomaterials-13-01326-f006:**
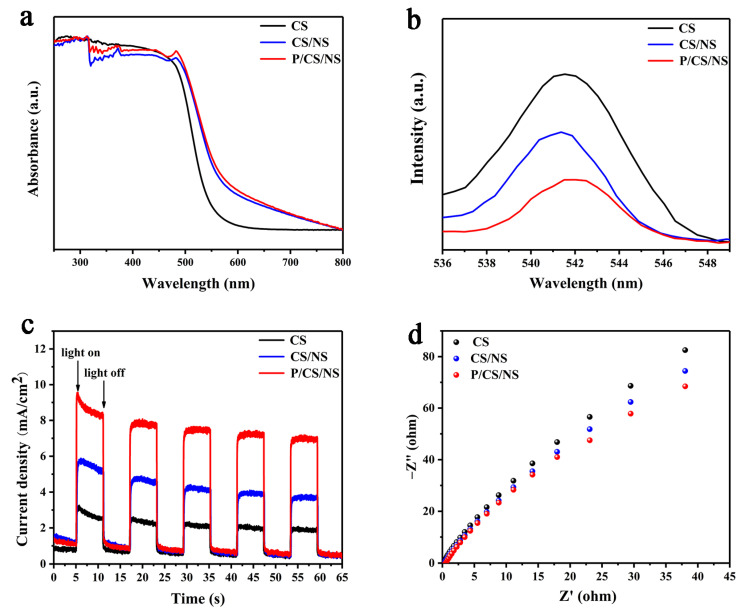
(**a**) UV-vis diffuse reflectance spectroscopy (DRS); (**b**) Photoluminescence spectra (PL); (**c**) Transient photocurrent response; (**d**) Electrochemical impedance spectroscopy (EIS) of CdS, CdS/NiS, and Pd/CdS/NiS.

**Figure 7 nanomaterials-13-01326-f007:**
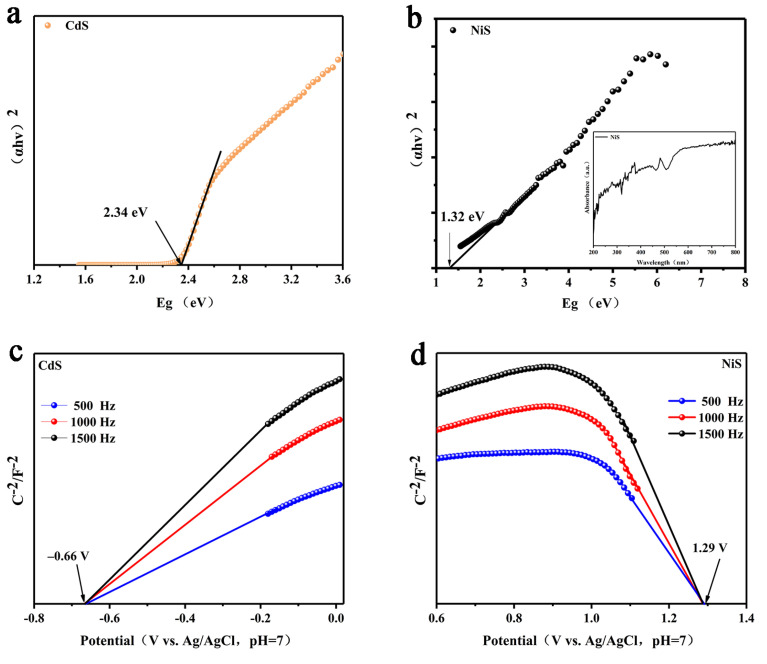
Tauc plots of (**a**) CdS; (**b**) NiS; Mott–Schottky plots of (**c**) CdS; (**d**) NiS (V vs. Ag/AgCl, pH = 7).

**Figure 8 nanomaterials-13-01326-f008:**
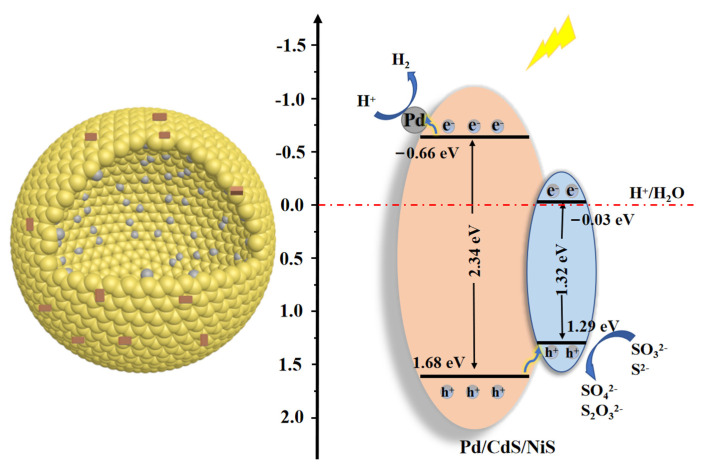
Schematic diagram of the photocatalytic reaction mechanism of the Pd/CdS/NiS.

## Data Availability

All of the relevant data are included in this published article.

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
