# Peer review of "Hollow Spherical Pd/CdS/NiS with Carrier Spatial Separation for Photocatalytic Hydrogen Generation"

_nanomaterials, 2023, doi:10.3390/nano13081326_

Round 1
Reviewer 1 Report
This paper describes the use of silica spheres as structural templates to grow a series of multilayer catalyst structures in the order of Pd/CdS/NiS with a thick photoactive CdS layer separating and interacting with the two other co-catalysts decorated on its different surfaces. There has been other work of a similar sort in the past, but the Pd separation from NiS may be a advance that could be of interest to photocatalyst readers. There several uncertainties in how well the actual materials match the proposed ideal models of the multilayer structure (or its composition) that could use additional thought and analysis.
In the figure 2 caption or graph, identify which reference structure is shown. Is there any direct evidence for Pd or NiS in the XRD?
In the main text, please use the full chemical names "CdS" and "NiS" versus the CS and NS shorthand - it may not be needed even in the chemical structure names as Pd/CdS/NiS is still a pretty short but much more readable core-shell structure designation than P/CS/NS.
The material used to get XPS results in Fig. 4 should be better defined. Was it a finely ground sample? If the answer is no, then how woudl XPS see both Ni and Pd signals as they are separated by a thick CdS barrier that could not show both XPS from materials on both sides of CdS.
One issue to clarify is some quantitative metals analysis on these different samples. There are conclusions made about the value of a certain chemical composition - 0.3% NiS and 0.7% Pd, but there no quantitative chemical analysis to verify that the reaction conditions resulted in different isolated core-shell chemical compositions. If the best performing sample was analyzed for Pd/Cd/Ni content, this would be a very useful addition to this article and likely improved its validity to photocatalyst researchers.
The legend in Figure 5 seems incomplete. Is the data in (a) with varying NiS using a single Pd content sample (amount not stated)? Same issue for (b) as it does not define the NiS amount used with the varying Pd content. (c) and (d) do not state the actual sample used or define what the purple data refers to. Are the units in Fig. 5a correct in saying they were scaled to catalyst mass in grams? From the other figures like 5b and 5e, it seems like the samples yield ~100 micromoles/hr without a mass normalization?
Figure 6 legend/captions do not clearly state which P/CS/NS composition is being studied and represented by the data shown.
The first sentence at the top of page 10 in an incomplete sentence.
Please review the text and limit the use of unnecessary prepositions that are used in nearly every 4th sentence, including "moreover", "meanwhile", "therefore", "first", subsequently, remarkably, furthermore, etc. as these words frequently do not add new information to the sentence versus just making the reader stop at the comma after the word.
Reviewer 2 Report
In this work, authors presented spherical Pd/CdS/NiS for photocatalytic Hydrogen generation. The results are just ok and I have several issues before making any decision.
- The idea of making hollow heterostructure with all these three components is not new and there are many reports on the same materials for the same applications of Hydrogen production. The authors should have highlighted the importance of this work, how this work is different than others and what is new achievement in this work.
- The figure 2 does not show any major difference in all three samples, why? There must be some extra peaks related to NS and CS? I cannot see any big difference in all samples.
- Can authors show line/point scan images of EDX rather than mapping to see the actual location of Ni, Pd in figure 3.
- They should compare their results with already published results of the same material in the form of a table, it is necessary.
- What is the band gap of heterostructure material? They presented BG of CDS and NiS individually in figure 7, but I want to see how the band gap changes when they have a composite structure.
- The discussion part is very poor and there are some works which reported the importance of oxygen vacancies in catalytic reactions, such as for CeO2 and Co3O4 as mentioned in these reports. https://doi.org/10.1039/C4RA03024A, https://doi.org/10.1155/2014/902730. The authors need to cite at least these reports.
- The English level of this work is very poor, and it need significant changes. There are many grammar and typo errors in this work, and they should be eliminated in the next version.
Reviewer 3 Report
The manuscript by X. Wang et al., is about hollow Pd/CdS/NiS hetero-nanoparticles towards photocataytic hydrogen evolution. With such an assembly they want to address and overcome the excess charge generated by light absorption on photocatalyst's surface. It was shown that the CdS helps to trap photo-induced holes. Moreover, the specific core-shell structure improves charge carrier separation. The study is interesting. However, the presentation as well as discussions need further details and corrections.
The authors often compare the achieved yield with the single-component CdS photocatalyst (for example, in the abstract). I think it would be appropriate if the yields were compared with the hollow spherical shape multi-component photocatalysts since there are already a lot of published data on this. In this regard, the following review paper might be useful. Catalysts 2022, 12(11), 1316
Similarly, in the Introduction section, the authors must emphasize better the novelty of their study in comparison with other studies already reported on hollow and hetero-nanoparticles containing Pd, CdS, and NiS nanoparticles (at least one of them).
In section 2.2, the authors mention about the cycling tests of the obtained photocatalysts. First of all, how the photocatalysts were treated for the recycling tests is missing. Moreover, the slight increase in the yield after 3rd cycle, which is ascribed as the formation of PdS. However, it means after saturation of all Pd ions, the reaction environment will be contaminated by the residual S. Can you elaborate on such issues further? Also, if the cycling test was performed continuously without refreshing the photocatalysts. This only shows the robustness of the photocatalysts in solution and during the photocatalytic process. However, it does not necessarily mean that the photocatalysts, particularly the surface-loaded nanoparticles, are stable enough for the washing/recycling process. Maybe the authors can add more in this regard.
The authors mention 3804 micromol/g/h H2 production by the formed photocatalyst. However, they put this data in the supplementary material. Instead, they show in Fig 5, the data with 400 micromol/g/mol H2 production. This should be better organized. The difference between these two yields must be clearly stated.
The conclusion is not on the basis of the obtained results but mainly consists of some general comments. It should be improved.
The manuscript contains many back and force, repetitive statements, very general assumptions, and also grammatical errors. I suggest the authors carefully read the text and eliminate the existing mistakes.
Reviewer 4 Report
The manuscript "Hollow Spherical Pd/CdS/NiS with Carrier Spatial Separation for Photocatalytic Hydrogen Generation" is very interesting and can be accepted after addressing the following comments.
1. The authors are missing experimental section, experimental protocol is there in electronic supplementary information, should Section 2 is Experiments and Section 3 is Results Discussion, change the according journal guidelines.
2. Figure 2 need to change X-axis labeling 2Ï´ (deg.) and in addition XRD should measures NiS further compare the other photocatalyst which is Pristine CdS, NiS, CS/NS, p/CS/NS and also index the NiS planes at corresponding 2Ï´ Values. Better to understand the readers, cite the Journal of Nanostructure in Chemistry volume 12, pages179–191 (2022), https://doi.org/10.1016/j.materresbull.2018.01.043
3. Figure 3 should be changing the center all figures should be follows as journal template, figure 3(b-i) labeling color change from white color to black color. For better visibility and understanding for readers?
4.in supplementary information, Label as Figure S1, Figure S2, and Figure S3, and should maintain the font size uniform. Figure S3 missing x-axis name concentration of Pd or Pd loading.
5. The authors should be inserting the photography setup for the photocatalytic hydrogen generation Exponential set up. And apparent quantum efficiency values are missing in abstract.
6. The authors should carefully check throughout the manuscript spacing of words for Example four hours change to 4 h, 0.28% change 0.28 % like this. Authors should site the https://doi.org/10.1016/j.ijhydene.2023.01.059, https://doi.org/10.1021/acsaem.9b00790
7. The authors should carefully check grammatical sentences in some places are inappropriate, spacing of the percentage and references suffix and prefix.
8. Figure 5 Y-axis label changes (a) y-axis Labelle from H2 Evolution (µmol h-1g-1cat) to rate of H2 (µmol h-1g-1), X-axis Concentration of CS/NS (wt %) . (b and d) y-axis Labelle from H2 production (µmol) to Volume of H2 (µmol g-1), Time (min) to Time (min.) Figure 5 (c) y-labeling AQY(%) to AQY (%).
9. Figure 6(b) y-axis labeling should be changes from Intensity (a.u) to Intensity (a.u.)
10. Abstract and conclusion should be more informatively with concise and insert few remarkable values such stability and APQY ?
Round 2
Reviewer 1 Report
The authors have improved the clarity of their paper though revisions requested by the prior reviews. Overall, the work appears carefully performed, adequately characterized, and shows appreciable H2 evolution enhancements with the multilayered catalyst structure. There are still some minor grammatical errors and changes in font type/size in several places in the manuscript.
Author Response
1.The authors have improved the clarity of their paper though revisions requested by the prior reviews. Overall, the work appears carefully performed, adequately characterized, and shows appreciable H2 evolution enhancements with the multilayered catalyst structure. There are still some minor grammatical errors and changes in font type/size in several places in the manuscript.
Response: Thank you for your valuable suggestions. We checked for grammatical errors and made corrections. Since the fonts may have changed during the resubmission process, we downloaded the latest version of the manuscript and made uniform changes to the fonts in it.
Reviewer 3 Report
The authors have addressed the reviewers' concerns. The manuscript can be accepted for publication.
Some references (i.e., 20, 21, 36, 37, 42 ) are not appropriately cited (the issue and page numbers must be mentioned) . This must be corrected before the final acceptance by the journal.
Author Response
The authors have addressed the reviewers' concerns. The manuscript can be accepted for publication.
1. Some references (i.e., 20, 21, 36, 37, 42 ) are not appropriately cited (the issue and page numbers must be mentioned) . This must be corrected before the final acceptance by the journal.
Response: Thank you for your valuable suggestions. We checked the literature and corrected the literature that was not properly cited (e.g., 12, 16, 20, 21, 36, 37, 42, 43, 44, 50, 52).
Reviewer 4 Report
The authors are well-improved Revision version, the same format accepted
Author Response
The authors are well-improved Revision version, the same format accepted
Response: Thank you for your valuable suggestions. We have revised the manuscript in accordance with the reviewers' comments.